# Perceived barriers and enablers influencing physical activity in heart failure: A qualitative one-to-one interview study

Aliya Amirova[1]*, Rebecca Lucas[2], Martin R. Cowie[3,4], Mark Haddad[5]

**1** Health Psychology Section, Institute of Psychiatry, Psychology and Neuroscience, King's College London, Guy's Hospital, London, United Kingdom, **2** St Raphael's Hospice Hospital & Health Care, Sutton, United Kingdom, **3** Royal Brompton Hospital, London, United Kingdom, **4** School of Cardiovascular Medicine & Sciences, Faculty of Life Sciences & Medicine, King's College London, London, United Kingdom, **5** Health Services Research and Management, School of Health Sciences, City University of London, London, United Kingdom

* aliya.1.amirova@kcl.ac.uk

**Data Availability Statement:** Qualitative data underpinning the manuscript cannot be shared publicly because of the nature of the data and ethics committee agreement (Health Research

## Abstract

In heart failure (HF), increased physical activity is associated with improved quality of life, reduced hospitalisation, and increased longevity and is an important aim of treatment. However, physical activity levels in individuals living with HF are typically extremely low. This qualitative study with one-to-one interviews systematically explores perceived clinical, environmental, and psychosocial barriers and enablers in older adults (≥70 years old) living with HF. Semi-structured interviews (N = 16) based on the Theoretical Domains Framework elicited 39 belief statements describing the barriers and enablers to physical activity. Theoretical domains containing these beliefs and corresponding constructs that were both pervasive and common were deemed most relevant. These were: concerns about physical activity (Beliefs about Consequences), self-efficacy (Beliefs about Capabilities), social support (Social Influences), major health event (Environmental Context and Resources), goal behavioural (Goal), action planning (Behavioural Regulation). This work extends the limited research on the modifiable barriers and enablers for physical activity participation by individuals living with HF. The research findings provide insights for cardiologists, HF-specialist nurses, and physiotherapists to help co-design and deliver a physical activity intervention more likely to be effective for individuals living with HF.

## Introduction

Heart failure (HF) is a syndrome triggered by underlying cardiac dysfunction that affects the efficiency with which the heart pumps blood around the body [1]. It is characterised by signs of volume overload, which may include peripheral oedema and pulmonary rales, and symptoms of breathlessness (dyspnoea), effort intolerance and fatigue. The extent to which these symptoms limit physical activity can be categorised using New York Heart Association (NYHA) classification (class I–no limitations; Class II–symptoms on ordinary exertion; Class

Authority. Cambridge Central Research Ethics Committee, REC reference: 17/EE/0183). Participants' approved consent specifically states that potentially identifiable participant data will not be shared. Given the nature of the qualitative transcripts and data analysis, although information on names, clinical team names and family members' names was retracted, the transcripts still contain detailed information that can be traced back to single individuals even in the aggregate form, given the study context. A point of contact: NRESCommittee.EastofEngland-Cambridge Central@nhs.net.

**Funding:** AA received City University of London PhD studentship. funder: City University of London The funders had no role in study design, data collection and analysis, decision to publish, or preparation of the manuscript.

**Competing interests:** The authors have declared that no competing interests exist.

III–symptoms on less than ordinary exertion; and Class IV–symptoms at rest) [2]. Traditionally, two main "types" of HF exist, heart failure with reduced left ventricular ejection fraction (HFrEF) and heart failure with preserved LVEF (HFpEF) [1]. HF is a global health problem, affecting an estimated 64 million individuals: around 1% to 2% of the general adult population [3]. Despite medical advances, HF is a substantial cause of morbidity and mortality [4]; 5 and 10-year survival have been identified by recent meta-analysis as 57% and 35%, respectively [5].

Physical activity is associated with improved quality of life [6–8], reduced hospitalisation [7], and increased longevity [9]. A physically active lifestyle, therefore, is a component of the recommended treatment [1]. However, levels of engagement are considerably lower among individuals with HF than in the general population [10]. Promoting physical activity is particularly challenging in the context of HF, owing to the complexity underlying the behaviour [11] and the many challenges individuals living with HF may face [12]. A systematic review of the barriers and enablers of physical activity among this patient group [13] indicated that a complex relationship between physical health status and support and encouragement from professionals and social sources are relevant and merit further investigation.

Several of the barriers to attending CR are speculated to exist at the healthcare system and broader socio-economic level (such as availability of cardiac rehabilitation programmes and exercise programmes, transportation issues, low referral rates, and limited knowledge of available programmes), as outlined in a consensus paper on adherence to exercise programmes and its barriers [12]. A recent systematic review identified that older age, depression and low LVEF are considerable barriers to everyday physical activity engagement in HF patients [14]. There is moderate evidence in support of the following modifiable barriers–symptom distress, negative emotional response towards physical activity–and modifiable enablers–social support, self-efficacy, and positive attitude towards physical activity. However, research into modifiable barriers and enablers is limited [14].

While the beliefs about symptoms and the course of the disease held by individuals living with HF have been previously described [15], less is known about how to promote recommended lifestyle changes, including increased physical activity. There is emerging evidence that physical activity interventions addressing behaviour change [16] and interventions based on a behaviour change theory [17] are potentially promising for promoting physical activity in HF. An explicit understanding of how behaviour is enacted is recommended by the Medical Research Council Guidelines for the development of complex interventions because this can enable understanding of the mechanisms that underpin change and so guide the choice of theory, thus informing the development and delivery of future interventions [11, 18, 19].

Currently, existing interventions have a limited and variable impact on increasing physical activity in older adults living with HF (9). Only one intervention–REACH-HF [20]–was developed adhering to the Medical Research Council guidelines [11] and evaluated a theory-based intervention targeting self-care behaviours, including physical activity engagement [20]. The REACH-HF intervention achieved clinically meaningful change in the quality of life and is estimated to be cost-effective in increasing the number of quality-adjusted life years, QALY [21]. However, the efficacy of the REACH-HF intervention in increasing physical activity in HF was not supported [22]. The REACH-HF intervention was developed using the Intervention Mapping Framework [23]. This study, which also follows the MRC guidelines, adopts alternative methods used in research informing intervention development Theoretical Domains Framework [24]. The use of TDF offers the following advantages over the IMF. The TDF offers means to systematise and structure qualitative analysis. Theoretical Domains Framework [25] is a tool developed through an international collaborative effort. It systematically describes domains and constructs which influence behaviour under investigation [24]. TDF summarises constructs of existing behaviour change theories into 14 domains, such as

Knowledge, Skills, Social/Professional Role and Identity, Beliefs about Capabilities, Optimism, Beliefs about Consequences, Reinforcement, Intentions, Goals; Memory, Attention and Decision Processes; Environmental Context and Resources; Social influences; Emotion; and Behavioural Regulation. Domains. TDF has been widely used in research [24], including research on physical activity in healthy adults [26, 27]. It has been suggested that TDF-based semi-structured interview helps in eliciting a greater number of relevant barriers and enablers, in contrast to unstructured interviews or less structured interviews which are likely to result in the identification of only some, usually the most salient, barriers and enablers [28]. Thus, TDF-based semi-structured interviews are expected to facilitate the search for a broader range of domains that are perceived as relevant to physical activity by individuals living with HF.

Additionally, in contrast to REACH-HF, the present study focuses on physical activity alone. A meta-analysis of behaviour change interventions suggests that simultaneously addressing several behaviours is not beneficial in achieving a positive change in the target behaviour [29].

This study expands the understanding of the barriers and enablers to physical activity in older adults (≥70 years old) living with HF. The present qualitative study with one-to-one interviews systematically explores perceived clinical, environmental, and psychosocial barriers and enablers in older adults living with HF.

## Methods

We followed the Standards for reporting qualitative research in conducting and describing this study [30].

### Design

One-to-one interviews guided by a TDF-based interview schedule were carried out.

### Setting and recruitment

Participants were recruited from outpatient cardiology clinics at the Royal Brompton & Harefield NHS Foundation Trust, UK. Individuals with HF who meet the inclusion criteria were identified by a member of the clinical team and asked if they were interested in the study. Those who expressed an interest were introduced to the researcher (AA). Each participant was provided with a participant information sheet (PIS) and an informed consent form (ICF). The researcher described the study aims, objectives, and procedure in more detail and answered participants' questions about the study. Individuals who expressed an interest in participating in the study were given an option to consider their participation over 24 hours. Those who decided to take part were asked to sign the ICF.

### Inclusion and exclusion criteria

Older adults (≥70 years old) diagnosed with HF according to the diagnosis criteria outlined by the contemporaneous European Society of Cardiology guidelines [31] were recruited to take part in this study. Only clinically stable HF patients were recruited (i.e., someone who has not experienced a change in their condition's severity, New York Heart Association (NYHA) class, or medical regimen in the past three months). The clinical assessment of the stable condition was carried out by a health professional at the recruitment site. To take part in the study, participants were required to be able to provide informed consent and to converse in English. Individuals with uncontrolled angina or symptoms even at rest (NYHA class IV) and those who were recommended to avoid exercise or any moderate or strenuous physical activity by a healthcare professional were not invited to take part in the study.

## Sampling strategy

The criterion sampling strategy has been designed to reflect the diversity and breadth of this population within pragmatic limits (i.e., ethnicity, sex, and NYHA class).

## Ethical issues pertaining to human subjects

The design and the conduct of this research were approved by the Health Research Authority. The ethical approval was received from the East of England–Cambridge Central Research Ethics Committee (REC reference: 17/EE/0183).

## Data collection methods

Age, sex, level of education, ethnicity, marital and occupational status were recorded using self-reports. Information on the duration of HF diagnosis, comorbidities, NYHA class, left ventricular ejection fraction (LVEF) (%) at the most recent clinical assessment, medication, and the frequency of hospitalisation in the past year was extracted from clinical records. Comorbidities were additionally assessed using a standard self-report checklist [32]. The medical records data extracts were matched to the participant identification number and entered into an Excel file.

## Procedure

After providing informed consent, participants took part in a single one-to-one interview. The researcher arranged for the interview to be conducted at a convenient time and place suited to the participant. Participants were offered an option to be interviewed at City, University of London premises, in a suitable room at the clinic (e.g., vacant consultancy room), research rooms available at the clinic, at the participant's home, or via telephone. All participants opted for the research room or clinic room at the hospital or for a telephone interview.

## Data collection instruments and technologies

A flexible interview schedule was developed. The schedule was designed to elicit the description of everyday physical activity. It then explored how physical activity has changed since the time of HF diagnosis to assist the participant in expressing beliefs that are relevant to HF. The schedule was then followed with a TDF-informed interview schedule. The feedback on the content and structure of the interview schedule was received from six health services researchers with expertise in TDF-informed research and/or health research management, a cardiologist, a HF-specialist nurse, and two individuals diagnosed with HF, who are members of the Patient-Participant Committee at the Public Involvement in Research Forum of the Cardiovascular Biomedical Research Unit at the Royal Brompton and Harefield NHS Foundation Trust. The interview schedule was amended following this feedback. One individual with HF participated in a pilot interview and provided feedback on the structure and length of the interview.

## Data processing

With the participants' consent, interviews were audio-recorded and transcribed verbatim. The audio recordings and the transcripts were pseudonymised by assigning a participant identification number to each interview. The analysis was facilitated using NVivo 12 software.

## Data analysis

We followed the five steps recommended for conducting data analysis of TDF-based interviews [24].

**Coding the interview transcripts.** A line-by-line analysis of each transcript involved categorising monothematic parses of text referred to as 'quotes' (e.g., phrase, sentence, a collection of sentences conveying a single meaning) into TDF domains. For each domain, the quotes with a shared underlying meaning were then summarised into belief statements.

**Inter-rater reliability of coding and coding scheme.** Three authors independently annotated a proportion of interview transcripts. The inter-rater reliability (Krippendorff alpha) was estimated [33]. A Krippendorff alpha of $\alpha \geq 0.80$ was considered as an indication of high inter-rater agreement among the three authors. A coding scheme was developed based on the initial data analysis, which then guided the analysis of the remaining transcripts.

**Generating specific belief statements.** Specific belief statements about the barriers and enablers of physical activity were generated from the quotes. Specific belief statements are defined as a collection of responses with a similar underlying theme that suggests a problem and/or influence of the belief on the target behaviour [28]. Strong evidence for a belief affecting the behaviour had to be present in each interview transcript for it to be coded as a quote. The frequencies of quotes supporting a belief statement (number of quotes, k) and the number of participants (n) were calculated.

**Generating causal belief statements.** Each transcript was assessed on the presence of lexico-syntactic patterns used to infer causality in natural language. The TDF domains and constructs that were linked, as evidenced by these patterns, were noted as being related to one another.

The following types of lexico-syntactic patterns inferring causality were extracted from the transcripts: 1) A causal statement about the event/experience/state indicated by the phrases such as 'because', 'since', 'as a result of', 'this led to', 'due to', (syntactic causal structures; [34]; 2) A causal statement indicated by the use of causal verbs (e.g. 'resulted in', 'decreased', 'increased', 'caused', 'made'); 3)A conditioning statement indicated by a semantic structure such as if-then and when-then. For example, 'since I had a surgery, I became more confident in exercising'; 4) A counterfactual statement [35] which include statements supporting reasoning about consequences of hypothetical but feasible situations or events that are contrary to the actual situation or event [36].

The descriptions of perceived cause-effect relationships between TDF domains and physical activity were extracted from the transcripts and used in building a graph that represents this causal structure.

**Identifying relevant theoretical domains.** The belief statements were categorised according to TDF. The relevance of the domains was evaluated by three authors.

*Quantification.* Domains were judged as likely to be relevant for changing physical activity in HF if they were either relatively pervasive (i.e., high number of quotes, k) or common (expressed by a large proportion of participants, n), or both. For a belief to be considered relevant, it had to be expressed by more than two participants at least. If a specific belief had a low frequency (number of quotes, k) but was mentioned by all participants, it was identified as relevant to physical activity.

*Differences between transcripts.* From self-reported within transcripts description of everyday physical activity, participants were categorised into (i) sedentary individuals and those performing (ii) moderate physical activity and (iii) vagarious physical activity at least once a week. The beliefs that were shared by all physically active individuals but not shared by sedentary individuals were judged as relevant.

**Mapping specific beliefs onto theoretical constructs.** Belief statements were mapped onto theoretical constructs. The constructs were chosen on the basis of how closely they described each belief statement in a single term. Standardised definitions of the constructs as described by TDF [24], the APA Psychology Dictionary [37] were adopted. *Environmental*

*Context and Resources* constructs were adopted from HF guidelines [1]. This has been done to situate the findings of this study in the vast body of literature on behaviour change and HF.

### Techniques to enhance trustworthiness

The first interview was independently annotated using the TDF by three authors. The proportion of the quotes that the coders agreed were as follows: author 1 and author 2 (50.90%); author 1 and author 3 (30.43%); author 2 and author 3 (32.30%). The Krippendorff alpha–a measure of inter-rater reliability–was 0.69, 95% CI: [0.63; 0.75], with a 61% chance of falling below 0.70 if an entire sample of the interviews would be coded. Since a sufficient level of agreement ($\alpha \geq 80$) was not reached, another two interview transcripts (comprising 242 coded quotes in total) were annotated using a coding scheme that was developed as the result of independent data analysis of the first transcript and discussion among three authors.

The proportion of quotes that the authors agreed on was as follows: author 1 and author 2 (81.67%); author 1 and author 3 (71.67%); author 2 and author 3 (63.33%). The Krippendorff alpha was 0.797, 95% CI: [0.72; 0.87], with a 0.5% chance of alpha falling below 0.70 if it was estimated from the entire sample. Any new disagreements were discussed. The new coding scheme was developed based on the analysis of all three interview transcripts and disagreement discussion. The remaining 13 interviews (755 quotes in total) were annotated by the first author using this coding scheme.

## Results

### Recruitment

The participant recruitment is described in Fig 1. One-to-one interviews were conducted face-to-face in a research room available at RBHT (n = 6), a vacant consultancy room (n = 6) and via phone (n = 4). All interviews were audio-recorded and transcribed verbatim. Interviews' duration ranged between 15 to 85 minutes (mean = 41.24, SD = 20.97).

### Participant characteristics

Participants' characteristics are reported in Table 1. The final sample included 16 participants of a mean age of 79.19 (SD = 5.15), four of whom were women, and 12 were men.

Sedentary behaviour (n = 1): Participant 5 was mostly sedentary, however used an exercise step once a week as prescribed by nurse and walked indoors.

Moderate physical activity (n = 15): All but Participant 5 walked daily. Two participants walked for at least 30 minutes as part of everyday activities of daily living (Participants 3 and 4). Other participants walked for leisure in addition to activities of daily living at least once a week.

Vigorous physical activity (n = 8): Three participants had an exercise routine (Participants 7, 11, 8); two attended a gym (Participants 2 and 1), one–a sports club (bowls game; Participant 15), one–an aqua-aerobics class (Participant 1), and one participant exercised regularly using a rowing machine (Participant 16).

### Findings

A total of 39 belief statements were produced from 16 transcripts. The corresponding constructs and the TDF domains, the number of quotes supporting them, and the number of participants expressing these beliefs are described in Table 2. The belief statements were mapped onto constructs and domains specified by TDF (12). The largest proportion of belief statements were coded as *Environmental Context and Resources*. The following domains were also

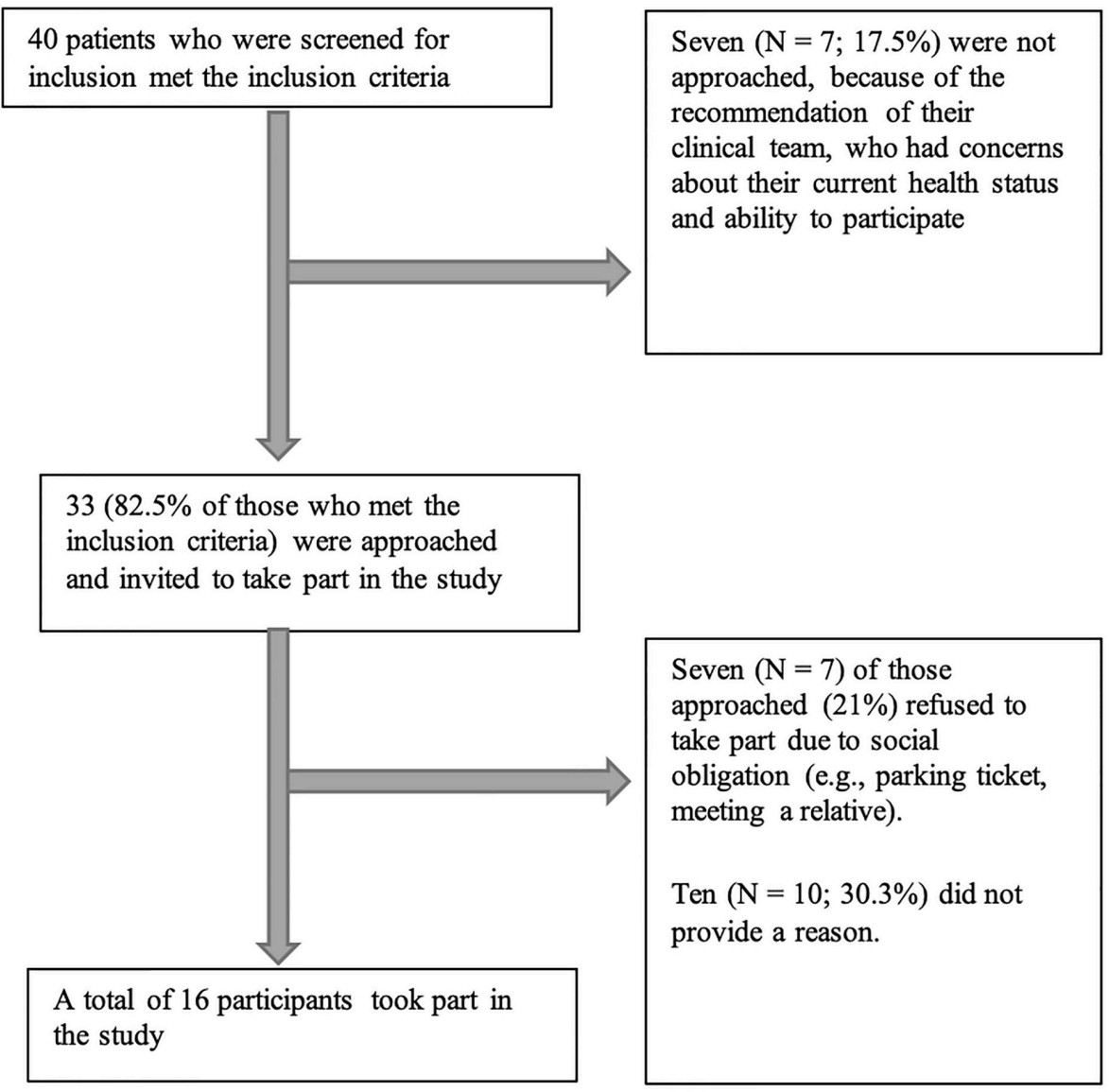

**Fig 1. Participant recruitment.**

judged to be relevant to physical activity based on the number of quotes: *Beliefs about Consequences*, *Goal*, *Social Influences*, *Beliefs about Capabilities*, and *Behavioural Regulation*.

The symptom of *breathlessness (i.e., dyspnoea)* reduced ability and damaged confidence (*Beliefs about Capabilities*); it led to persistent anticipation of negative outcomes of physical activity (*Beliefs about Consequences*). Due to this multifaceted influence of *dyspnoea* on physical activity, it is identified as a major barrier to physical activity in HF.

## Causal belief statements

The causal belief statements summarising the perceived links between domains that were shared by at least three participants are presented in Table 3. These were summarised into a causal graph (Fig 2). All listed domains were causally linked to physical activity.

**Table 1. Participant characteristics.**

| Demographic Characteristics | | Clinical Characteristics | |
|---|---|---|---|
| Age (mean, SD) | 79.19 (5.15) | Number of comorbidities (sample mean, SD) | 4.88 (2.39) |
| | | *Atrial fibrillation* | 6 (37.5%) |
| | | *Rheumatoid arthritis* | 6 (37.5% |
| | | *Myocardial infarction* | 4 (25%) |
| | | *Aortic stenosis* | 3 (18.75%) |
| | | *Pulmonary hypertension* | 3 (18.75%) |
| | | *Mitral regurgitation* | 3 (18.75%) |
| | | *Diabetes Meletus type II* | 2 (12.5%) |
| | | Cardiac implantable device, n (%) | 4 (25%) |
| Sex, n (%) | | LVEF, % (mean, SD) | 33 (14) |
| *Male* | 12 (75%) | NYHA Class I | 1 (6.25%) |
| *Female* | 4 (25%) | NYHA Class II | 10 (62.5%) |
| Education | | NYHA Class III | 5 (31.25%) |
| *Postgraduate degree, n (%)* | 1 (6.25%) | Hospitalisation frequency in the past year | |
| *University degree, n (%)* | 3 (6.25%) | *None, n (%)* | 7 (43.75%) |
| *GCSE/A-levels or equivalent, n (%)* | 10 (56.25%) | *Once, n (%)* | 7 (43.75%) |
| *No formal education, n (%)* | 2 (12.50%) | *Twice, n (%)* | 1 (6.25%) |
| Ethnicity | | *Three times, n (%)* | 1 (6.25%) |
| *British–White, n (%)* | 11 (68.75%) | Hospitalisation duration, days (mean, SD) | Median = 4, IQR: [0.75; 5.75], Range: [0; 23] |
| *British–Pakistani, n (%)* | 1 (6.25%) | | |
| *White–Irish, n (%)* | 1 (6.25%) | Pharmaceutical treatment | |
| *White Other, n (%)* | 1 (6.25%) | *Beta-blockers* | 12 (75%) |
| *Asian Other, n (%)* | 2 (12.50%) | *Diuretics* | 2 (12.2%) |
| | | *Mineralocorticoid receptor antagonists* | 8 (50%) |
| | | *Renin angiotensin system inhibitors* | 15 (93.75%) |
| | | Total number of medications (mean, SD) | 6.94 (SD = 2.17) |

## Discussion

This study identified and described perceived barriers and enablers to physical activity in older adults living with HF. The following TDF domains were identified as relevant: *Environmental Context and Resources*, *Beliefs about Capabilities*, *Goal*, *Behavioural Regulation*, *Beliefs about Consequences*, and *Social Influences*. The specific beliefs summarising the barriers and enablers to physical activity are myriad (n = 39).

The lack of self-efficacy (*Beliefs about Capabilities*) influenced by the heart condition, comorbidities, and HF symptoms is suggested to be a pervasive barrier to physical activity in HF. Similarly, previous qualitative evidence suggests that '*Changing Soma*' [38] due to age, comorbidities, and HF causes lack of perceived ability (i.e., 'negative beliefs about perceived ability' [39]. In this study, the extent to which self-efficacy determined the engagement in physical activity was defined by the degree of *dyspnoea* as well as comorbidity. In contrast to a previous qualitative study [38] suggesting that the change in the perception about oneself because of ageing' influenced physical activity engagement, in this study, ageing was described to influence perceived capability rather than self-concept. Self-efficacy in its turn promoted resourcefulness in *Behavioural Regulation* (i.e., pacing oneself in accordance with perceived ability) as indicated by the causal belief statements.

The local environment and exercise equipment and facilities (*Environmental Context and Resources*) were reported as both barriers and enablers. It was identified as an enabler if it has

**Table 2. The TDF domains, constituting belief statements, corresponding constructs and their relevance to physical activity in HF.**

| Theoretical Domains Framework and constituting belief statements | Exemplar quote | Construct | Barrier/ Enabler | Num. participants (N) | Num. quotes (k) |
|---|---|---|---|---|---|
| **Goal Domain** | | | | **15** | **109** |
| Engaging in physical activity is a priority for me | *"I feel I have to push myself. Erm. . .I resist sitting down and do nothing. [. . .] Well, you know, I've got to do it."* | Goal priority[2] | Enabler | 15 | 52 |
| I engage in physical activity to be able to get on with life without help from others | *'I want to be self-sufficient still, I don't want to rely on other people. So, staying fit is important to me.'* | Outcome goal[2] (extrinsic motivation[1]: functional independence) | Enabler | 9 | 23 |
| I have integrated an adequate amount of physical activity into my life | *'Now I will do it every day and if I can twice a day, to the point it feels good. I'm going to build up with it to the point where it feels good. [. . .] You take everything to the edge until it . . .and you know all exercise should begin to hurt and that's when you stop.'* | Behavioural goal[2]: Goal attainment* | Enabler | 7 | 18 |
| Engaging in physical activity is (not) a priority for me | *'Well. . .I know walking is a solitary experience, I spent a lot of time on my own. . .So, why would I go outside? This is how I look at it, why would I go outside on my own to walk around, when I am far happier being on my own in doors with a book.'* | Goal priority[2]/Goal conflict | Barrier | 5 | 8 |
| I already engage in as much physical activity as I am able to | *'So, this is what I do [summarising bowls, walking and exercise routine] and I think it suits me and I'm happy to do that I can't do more than that.'* | Behavioural goal[2]: capability-corresponding goal* (lack of intrinsic motivation[1]) | Barrier | 2 | 8 |
| **Environmental Context and Resources** | | | | **15** | **99** |
| My local environment limits me in engaging in physical activity (incline (hills); crowds; traffic; pollution) | *'Or I may have a giddy turn and I have to stop, so it is OK (to walk) where I live but when you are in London and there are people pushing and shoving all the while, it is not the ideal place to be.'* | Local environment[1] | Barrier | 8 | 20 |
| *Major event* | | | | | *20* |
| My physical activity levels decreased since health-related event | *'Why did I stop [walking long distances mentioned earlier in the interview]? [. . .] cause of an operation I had. . . one of the heart operations. . .they said to be careful the first month. . .and fundamentally, when you stop you don't start again. . .at my age* | Health-related event[2] (barrier) | Barrier | 7 | 11 |
| My physical activity levels increased since health-related event | *'So, the heart attack was a trigger, the catalyst for change [in participant's lifestyle which participant described earlier in the interview]'.* | Health-related event[2] (enabler) | Enabler | 2 | 4 |
| My physical activity levels decreased since a major life event | *I worked for 20 or 30 years; I think. I used to get pretty busy. . .I used to have a little office in a new building, and there will be a walk rail. . .all the way around. . .Quite a big square, which is undercover, and periodically, I would get up and walk around the corridors (laughs). And now I don't have that.* | Major life events[2] (barrier) | Barrier | 3 | 3 |
| My physical activity levels increased since a major life event | *'Certainly, since I have retired [I start4ed my exercise routine in the morning]. Getting up in the morning to go to work, I didn't have time to do it, when I was a commuter. It is certainly since I have been retired.'* | Major life event[2] (enabler) | Enabler | 1 | 2 |
| *Treatment* | | | | *7* | *17* |
| My HF treatment prevents me from engaging in physical activity | *'I don't [go for walks] because this water retention tablets make me very sick, made my stomach very upset. [..] Two, three steps. . .. I am afraid I will fall because of this medication. Maybe with this this. . .blood thinners, very strong medication. I think it is because of that.'* | HF treatment[4] | Barrier | 3 | 9 |

*(Continued)*

**Table 2.** (Continued)

| Theoretical Domains Framework and constituting belief statements | Exemplar quote | Construct | Barrier/ Enabler | Num. participants (N) | Num. quotes (k) |
|---|---|---|---|---|---|
| My HF treatment (i.e., medication) helps me in engaging in physical activity | 'I find it quicker and better to sit down and rest until it settles down, take my inhalers and then start again gently.' | HF treatment[4] | Enabler | 4 | 8 |
| *Implantable device* | | | | *4* | *13* |
| Having an implantable device reassures me when engaging in physical activity | 'And of course, since I had the ICD implanted it became much more relaxed. Because they know that these devices will recover you whatever the situation you are presented with. It is a reassurance factor; it is quite astounding. That psychological bit, you know that you are not doing yourself any harm [by exercising].' | Implantable device[4] (enabler) | Enabler | 4 | 9 |
| My implantable device can harm me if I engage in physical activity | 'Also, with my pacemaker…I have to be careful. I have to make sure I do not overdo it [physical activity]. You know?…they tell me…the more you do you use up the battery. This is my second one, I had it for six years. The first one I had for just over 3 years…I think you have to be careful about how much you do things.' | Implantable device[4] (barrier) | Barrier | 2 | 4 |
| Equipment (bike; treadmill) helps me in being active | 'I have this rowing machine in the corner, it's just the way it looks, a very small piece of furniture. It looks good, and it's very convenient to use.' | Equipment[4] | Enabler | 5 | 16 |
| Group programmes help me in being physically active | 'Actually, one thing that I did was going to cardiac rehab. To be honest, that was…it was like…' Oh God! I am too weak to do this', and they would say: 'No, you are not, you can walk up those 5 yards there, can't you?'… Ok, I can do that'. | Exercise-based group programmes[4] | Enabler | 5 | 8 |
| Facilities (e.g., local council) help me in being physically active | 'Being part of the club [bowls club], I know what days we play and when to come we just do it twice a week. Sometimes a bit more. | Facilities[4] | Enabler | 4 | 5 |
| **Social Influences Domain** | | | | **11** | **99** |
| I engage in physical activity because a health professional (e.g., GP, cardiologist, nurse) has advised me to do so | 'See, my consultant *** [cardiologist] told me I have to walk and move as much as I can. So, he told me I should stand up now and then and walk around the flat, so I do that.' | Health professional's advice[4] (enabler) | Enabler | 7 | 26 |
| People who are important to me encourage me to be physically active | 'When I have decided to go out or scheduled to go somewhere…it normally takes me, as I said up to 10:30 to get ready but if somebody says I am going to pick you up at nine. I actually can be ready by nine. […]My friends are all fit, and my partner, she plays golf and tennis. She is very fit. Again. that is a motivation for me, she walks about when we go out […] I keep up with her, except when it comes to hills, but normally we like to go away and do lots of things.' | Social support[3] (emotional, enabler) | Enabler | 7 | 15 |
| I would engage in physical activity if it involved being with others | 'Probably walking with somebody else, when you are walking on your own or me…when i am walking on my own, you have to be motivated to get yourself out of the chair but if I know a friend was coming around then yes. I would then go and I wouldn't think twice about it.' | Social Support (practical)/ Companionship* | Enabler | 5 | 12 |
| People who are important to me discourage me from engaging in physical activity | 'My daughter always tells me that I do too much. She was very annoyed when she came and saw that I cut the grass and tided the garden.' | Social support[3] (emotional, barrier) | Barrier | 5 | 11 |

*(Continued)*

**Table 2.** (Continued)

| Theoretical Domains Framework and constituting belief statements | Exemplar quote | Construct | Barrier/ Enabler | Num. participants (N) | Num. quotes (k) |
|---|---|---|---|---|---|
| Having a reassurance from a health professional that physical activity is safe encourage me to exercise | 'You look up to professional and you need that reassurance to say: 'what I am doing now, is not going to impact on my long-term health'; 'is it the right thing to do?' and if the answer comes back from a professional you trust, and I have implicit trust, explicit trust as well, in my consultant. . .then if she says: 'that's a good thing to do', then I will do it.' | Health professional's advice[4] Reassurance (Social support to reduce risk perception)* | Enabler | 3 | 9 |
| Making plans with others encourages me to engage in physical activity | 'I think if I didn't have friends and didn't schedule. . .I might have vegetated and sit a bit too much. . .it is important to have friends you can do things with. . ." | Social support (practical)[3] linked to behavioural regulation | Enabler | 4 | 7 |
| I rely on other people to perform physical activity | 'When I am with my daughter, I am less nervous and I am able to walk further. . . because she was there. . .and I walked very slowly but I did walk all the way and I was very pleased with myself, because I felt I have achieved something.' | Social support[3] (practical) | Barrier | 4 | 6 |
| I limit my physical activity because a health professional (e.g., GP, cardiologist, nurse) has advised me to not overdo it | 'Today I have done a bit of walking, and that's enough, you cannot overdo it. You tell your doctor that. . . they tell you off. . . they get upset if I don't do 22 minutes per day, and they get upset if I say: "I walked 45 minutes". Because they think some things might happen.' | Health professional's advice[4] (barrier) | Barrier | 2 | 5 |
| I am physically active because everyone I know closely is physically active | 'My family have always been active. I've always been active. So it's not something that happened suddenly.' | Social norm | Enabler | 3 | 4 |
| I would engage in physical activity with others if their level matched my capability | 'Well, let me say there is no one I could walk with, who would be a at my pace. . .and wouldn't be interested in walking half a mile. . .they DO walk but they would be interested in walking further and quicker than I do. So, I would be a liability to them.' | Social comparison[1] (linked to self-efficacy) | Barrier | 2 | 4 |
| **Beliefs about Consequences Domain** | | | | | **89** |
| *Positive outcome expectancy*[1] | | | *Enabler* | *11* | *52* |
| Physical activity improves my general health | 'Yes, it [walking] does it helps me to breathe better. It helps your blood to circulate better, which carries oxygen. . .I think it gives you a better feeling of well-being generally.' | Positive outcome expectancy[1] | Enabler | 13 | 36 |
| I engage in physical activity because it makes me feel more cheerful | 'I like to be happy and I only found . . .it is. . . if I do this [exercise] I remain happy! and you know it gives me. . .you know. Cheers me up.' | Positive outcome expectancy1 | Enabler | 6 | 12 |
| Physical activity improves the condition of my heart | 'I am not showing any symptoms of my heart condition and I think it is because I have been doing all these things [cycling using e-bike, aqua-aerobics and walking], When I walk more, I feel better, my circulation improves.' | Positive outcome expectancy1 | Enabler | 3 | 4 |
| *Negative outcome expectancy:*[1] | | | *Barrier* | *10* | |
| Physical activity brings on my symptoms (e. g. breathlessness) | '. . . sometimes if I hurry too much or going up a hill, I will get a bit short of breath' | Negative outcome expectancy1 | Barrier | 10 | 29 |
| Physical activities bring on my symptoms (e.g. tight chest; swollen legs; extreme fatigue) | 'I wouldn't like to keep on going when I stop because my chest is tightening up and my arms and my legs are getting heavy. . .and I am glad to be sitting down puffing.' | Negative outcome expectancy[1] | Barrier | 2 | 2 |

(*Continued*)

**Table 2.** (Continued)

| Theoretical Domains Framework and constituting belief statements | Exemplar quote | Construct | Barrier/ Enabler | Num. participants (N) | Num. quotes (k) |
|---|---|---|---|---|---|
| Physical activity Is dangerous because it puts my heart under strain | 'I want to go on. . .but why should I give myself a lot of pain. . .and put myself in danger, when stopping that would stop as well. . .if it kept on going I suppose I would be ringing 999 and would be taken off to a hospital.' | Risk perception[1] Concerns | Barrier | 4 | 6 |
| **Behavioural Regulation Domain** | | | | | **57** |
| I pace my physical activity to match my physical ability | 'I. . . sometimes if I hurry too much or going up a hill, I will get a bit short of breath. But I always. . .I pace myself more at home, which means if I'm in the middle of, say, cleaning or hoovering the house. I may have to sit down a bit more.' | Implementation intention[1] | Enabler | 10 | 30 |
| I have a physical activity routine I follow | 'I just wake up put my dressing gown on, go down the stairs, take my medication and do my exercises. It is just a routine. [. . .] But as a normal routine I just get up in the morning and do my exercises automatically.' | Habit | Enabler | 3 | 9 |
| I monitor the intensity and or duration of physical activity to make sure I do not overdo it | 'But I don't look at it [pedometer] until I feel I have done enough. 'Let's have a look at this.. oh well, it is over half an hour '. . .and then I stop. It is never more than 22 minutes.' | Self-monitoring[1] (downregulation driven by risk perception) * | Barrier | 2 | 6 |
| I know when and where I will engage in physical activity over the next week | 'Most days I walk a lot, everywhere. And when the schools come back, I will start swimming again once or twice a week.' | Planning[1] | Enabler | 3 | 4 |
| I monitor intensity and or duration of physical activity to make sure I do enough | 'Yeah, I have my Fitbit, you can see how many steps I do. [. . .] Fitbit encourages me to do more. [. . .] Because I have this information. [. . .] OK, 145 minutes. . . [. . .] This is last week. . .Altogether, 30013 steps, look! I did less [before Fitbit] . . .I still was doing it, but less. [. . .] I didn't have the record to see the results.' | Self-monitoring[1] (attainment)* | Enabler | 3 | 4 |
| When weather is bad, I engage in physical activity indoors | 'I walk even if it is raining, I walk in my flat for an hour and a half or something. If I can't go out, I walk inside every day. [. . .] Is it raining? Yes–then walk inside the flat, if it's not raining, I go out.' | Implementation Intention[1] | Enabler | 3 | 4 |

Note:

[1]. The definition and the construct term is adapted from APA dictionary;

[2]. The definition and the construct term is adapted from the TDF framework

[3]. The definition a is adapted from BCTTv1;

[4]. The definition and the term is adapted from NICE guidelines;

*The construct definition and term is not widely used and not empirically supported.

The preliminary names are noted to preserve the specificity of the belief statement content.

the following attributes: available at home, flexible (e.g., exertion levels to accommodate different physical capacity levels), safe, and aesthetically pleasing in its appearance. An intervention typology [40, 41] describes these features as essential for a successful nudging of the behaviour as functionality, availability and presentation, respectively. In HF, some tailoring to individual needs might be required (i.e., level of exertion being closely matched to their physical capacity and perceived ability).

The implantable device (*Environmental Context and Resources*) was a source of both–reassurance in, and worry about, physical activity. The NICE [1] and European Society of Cardiology [31] recommend low to moderate levels of activities in this population such as walking

**Table 3. The causal links described by at least three participants expressed in a belief statement.**

| The causal link and a belief statement | Participants | Illustrative quotes |
|---|---|---|
| I. **Dyspnoea → Beliefs about Capabilities**<br>I cannot engage in physical activity because I get out of breath.<br>I should not engage in physical activity if I experience dyspnoea (severe breathlessness).<br>Since I started experiencing breathlessness, I feel less able to walk. | Participant 6, 8, 13, 14 | Participant 6: 'Because I get breathless very quick and. . . We said about my blood pressure. . .there are lots of things, few things that I can't do now, some are related to age, and some are related to blood pressure.'<br>Participant 8: 'Oh, breathing! Then I shouldn't do it then. [. . .] I feel tightness (points at chest), and I feel that I'm not fit for running, so I shouldn't run, then I go fast walking and I found it difficult as well. So, no running; no fast walking. . .just normal walking.'<br>Participant 14: 'No, it is breathlessness. If it was not for breathlessness, I would walk more than I do. I used to walk more than I do now. Since I got this extreme breathlessness, I can't.' |
| II. **Environmental Context and Resources → Goal**<br>Group exercise programmes helped me to set suitable physical activity goals for myself. | Participant 1, 10, 15 | Participant 1: '. . .and in those days, they probably still do, you had you know rehab. . . and it involves doing exercise. . . in a gym. . .for half an hour or so, you know. . . structured exercise. . . and it was there [when] I first learnt: 'do as much as you feel comfortable with.' |
| III. **Environmental Context and Resources → Behavioural Regulation**<br>Monitoring devices (e.g., Fitbit) help me to monitor and regulate how much I engage in physical activity.<br>A major health-related event resulted in physical activity habit-breaking | Participant 8, 10, 16 | Participant 8: 'Interviewer: And before you got Fitbit did you do as much as you do now? Participant8: No. . .I did less. . .I still was doing it, but less. Especially I didn't have the record to see the results.'<br>Participant 10: 'I don't remember what it was, pacemaker probably. I don't remember. I remember being told to be careful [after a surgery] for a few months and not to do exercises as such. Once you get out of the habit, you . . .it is hard to start again. . .it is difficult to start again, because you forget.'<br>Participant 16: 'I tried to keep it going because you must have some sort of discipline to keep it going, otherwise you just stop. I don't have a plan, but I try to make sure that I don't miss too many.' |
| IV. **Social Influences → Behavioural Regulation**<br>Making plans with others encourage me to engage in physical activity.<br>I rely on others to plan and engage in physical activity. | Participants 2, 4, 5, 6 | Participant 2: 'Yes, I do. . .I think. . .when I have decided to go out or scheduled to go somewhere. . .it normally takes me, as I said, up to 10:30 to get ready, but if somebody says I am going to pick you up at nine. I actually can be ready by nine.'<br>Participant 4: 'Does planning your activities help you do more? Participant4: It would probably help me to do more, we certainly used to plan more, because we did it in conjunction with friends. . .and you have to plan if you are organising, but otherwise. . .'<br>Participant 5: 'My wife helps me to keep up with the schedule. Interviewer: Amazing! What does your wife say about these exercises or your activity in general? Participant5: she is the driving force of me doing all these, really. (laughs) She makes sure I do the exercise and the rest of it. . .'<br>Participant 6: 'As I am on my own, I would not make a plan. . .but if I was with other people, a friend and they exercised regularly, every Tuesday morning, and they asked me to join them.' |
| V. **Social Influences → Intention**<br>I intend to engage in physical activity only if I am with others<br>I intent to engage in physical activity being guided by the example of the people who are role models in my life. | Participants 2, 6, 7, 15 | Participant 2: 'Yeah it is really, I copy after them (smiles). . .Erm. . .I try to control my weight a bit. . .but. . .I have lost some recently. I should. . .again it is in the head, I got machines at home I could exercise on. . .. but when I am by myself, I don't really have the motivation to do it.'<br>Participant 6: 'If I had people, who were. . .going around on. . .erm. . .on. . .just walking really. . .or maybe the treadmill, you know people I know who are doing these things? I would, as a friend, I would join them, but I am not very good at going somewhere and getting on a treadmill and doing that for 10 or 15 minutes or whatever. . .and getting off and coming back home.'<br>Participant 7: '[. . .] And I've always been active. My parents have always, especially my mother, stressed the importance of being active, and to get out and do walking and everything. Yes. And I did it with my daughter. . . and my oldest, she does it with her daughter.'<br>Participant 15: 'Older friends. They are marvellous in so many ways, and they had to get on with life. When I got to their age. . .. I learnt from them: you got to do your best.' |

(*Continued*)

**Table 3.** (Continued)

| The causal link and a belief statement | Participants | Illustrative quotes |
|---|---|---|
| VI. **Beliefs about Capabilities →Beliefs about Consequences** I am aware of negative outcomes of physical activity (i.e., hospitalisation) that are part of having a comorbid condition. | Participants 2, 3, 4 | Participant 2: 'I have my Heart. . .heart problem, I have atrial fibrillation [. . .] And Erm. . .when I go to the gym, a lot of machines have this heart monitor thing. And sometimes when something goes off the sky. . .ooouh [tone of uncertainty], Shall I be doing that?' Participant 3: 'If I did more when I was quite asthmatic, by which I mean difficult to breath, it would make it worse.' |
| VII. **Beliefs about Capabilities → Behavioural regulation** I pace my physical activities accordingly to how capable and fit I fill at any given movement. | Participants 4, 6, 7 | Participant 4: 'I find it quicker and better to sit down and rest until it settles down [breathing], take my inhalers and then start again gently. I am guided by my body rather than to force my body to do thing.' Participant 7: 'I can do (get out of breath) at times during the day. If. I if I hurry too much. So, I've learned how to pace myself.' |
| VIII. **Beliefs about Consequences → Goal** I reduce the intensity and amount of physical activity I aim to engage in order to avoid negative outcomes, such as hospitalisation and worsening of my condition. | Participants 2, 3, 8 | Participant 3: 'Because, you know. . . today I have done a bit of walking, and that's enough, you cannot overdo it. You tell your doctor that. . . they tell you off. . . they get upset if I don't do 22 minutes per day, and they get upset if you say I walked 45 minutes because they think some things might happen.' Participant 2: 'And sometimes when something goes off the sky. . .ooouh [tone of uncertainty]. I [then] just slow down a little bit and start again. . .I try to keep my heart rate under 130.' Participant 8: '[. . .] I found . . . If I do this [walking and meeting goals on Fitbit], [then] I remain happy; and, you know, it gives me. . .you know, cheers me up, I don't have to. I don't have to be sad, jobless or workless or ill in front of my children. When I do this, I [then] can remain happy and smiley and things like that.' |
| IX. **Goal → Behavioural Regulation** I monitor my physical activity to make sure I do not overdo it. The goal to remain functionally independent pushes me to make plans for being physically active despite setbacks. | Participants 1, 2, 3 | Participant 3: 'I bought this wristwatch; it is like a watch, but it tells you how many steps you have done. Interviewer: Ah, right! Does that help you to do more? Participant3: Well, no, it helps me to stop, cause if I have overdone 22 minutes. But I don't look at it until I feel I have done enough. 'Let's have a look at this. Oh well, it is over half an hour '. . .and then I stop. It is never more than 22.' Participant 2: 'No, because I know I got to do it to stay where I am. I know it is going to give me a problem . . . I could use the station's lift. . .but It is important that I don't. . . but, you know, with what I got, and the atrial fibrillation as well, I mean, it could have been acceptable for me to sit in a chair in a nursing home, basically. . . OK? But I dread the thought of that. . . so, I keep going.' Participant 1: 'Well, I was trying not to be, I was dependent on my wife carrying me up. . . and daughter. . .carrying me up to the gym to do the aqua-aerobics. . . (and of course part of the boredom factor, if you like. . . that's engendered.) That's why I bought the e-bike. . . I was. . . at least I could I get around independently, without having to depend on my wife or daughter.' |

and daily activities, as well as soft aquarobics activities. Only sports of bodily collision and highly vigorous competitive sports are not recommended to this population group. Therefore, rightly so, participants regulated the type of activities they engaged in. However, several misconceptions surrounding their implantable device were still present. This included battery usage and wires imposing risk on the heart. As suggested by the review of the perceptions about cardiac implantable devices, education on the safety of the device would be helpful to individuals living with HF who were fitted with an implantable device.

Major health-related events, such as cardiac surgery, cancer surgery, heart attack, the first HF decompensation and subsequent hospitalisation, had a negative impact on physical activity for most of the participants (n = 7). These events were suggested to cause habit-breaking as indicated by the causal lexico-syntactic patterns' analysis. The habit-breaking follows a short period of clinically recommended rest (approx. three months). To put it in the words of Participant 10: "*I remember being told to be careful. Once you get out of the habit [post-surgery], it is hard to start again. It is difficult to start again because you forget.*" In addition, such events in

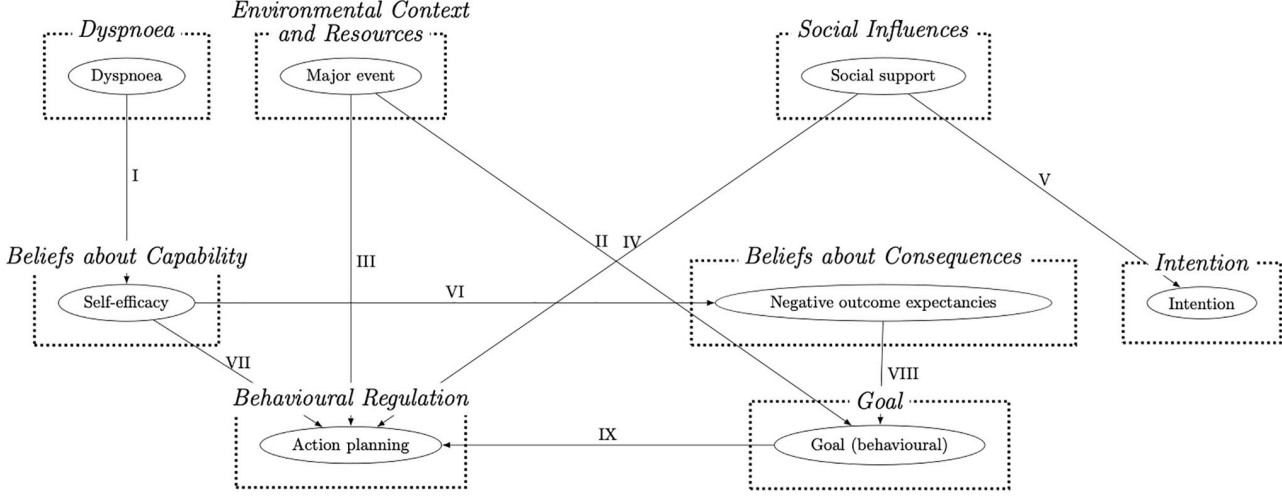

**Fig 2. The graph summarising causal links as supported by the lexico-syntactic patterns analysis.**

HF are associated with a drastic reduction in physical capacity. A significant life event–a crisis–is theorised to cause a shift in an individual's identity [42, 43]. The salience of the event may result in a radical shift in beliefs [42, 43]. This is especially likely given the associated risks of sedentary behaviour in HF, including reliving the crisis all over again.

*Behavioural Regulation* influenced physical activity in HF. These included implementation intentions–strategies to execute a goal-directed behaviour in the face of barriers such as bad weather, forgetfulness (i.e., routine), fluctuation of perceived ability (i.e., pacing of physical activities when feeling unwell) [44]. The following strategies that require focus and effort were also useful: action planning; and self-monitoring. These are consistent with: Self-regulation theory [45], and Dual process model of self-control [46]. *Behavioural Regulation* implies prolonged effortful control of behaviour and restriction, which may result in ego depletion–the loss of motivation, cognitive resources, and the focus on the behaviour [47]. Thus, making it an inconsistent enabler of the behaviour. Social and environmental resources such as social support and companionship (*Social Influences*) and monitoring devices (*Environmental Context and Resources*) were utilised to support *Behavioural Regulation* as indicated by the lexico-syntactic patterns' analysis. *Beliefs about Capabilities* were causally linked to *Behavioural Regulation* constructs such as implementation intentions. Clinical advice (enabler) and reassurance, Companionship, Social support (practical), Social support (emotional, barrier), Social support (emotional, enabler), Social comparison, Social support (practical), and Social modelling were all suggested to influence physical activity in HF. These *Social Influences* were explicitly causally linked to *Intention* and *Behavioural Regulation*, as evident by the lexico-syntactic patterns. Three participants who self-identified as active were inspired by the positive example set by individuals they viewed as role models. Participants who did not self-identify as active intended to exercise only when provided with social support (practical and emotional) by others. Negative *Beliefs about Consequences*, such as negative outcome expectancies (i.e., symptom onset and hospitalisation), were causally linked to a weakened behavioural goal (*Goal*). Participants were less likely to form a strong behavioural goal to engage in physical activity if they held negative outcome expectancies about the behaviour. Action planning was motivated by the positive outcome goal (*Goal*): to remain functionally independent. A negative goal targeted

at avoiding overdoing physical activity resulted in monitoring physical activity with an aim to reduce it.

Overall, the causal relationships between the identified barriers and enablers and described using lexico-syntactic patterns analysis in this study are consistent with Health Action Process Approach [48, 49]. The HAPA suggests that self-efficacy enables an individual to be resourceful in action planning and coping. This study supports the relevance of HAPA in explaining the driving forces of physical activity in HF, and thus it might be useful to employ this theory in developing physical activity interventions designed for individuals living with HF.

*Dyspnoea* is suggested a key barrier to physical activity in this study and an overarching theme spanning two TDF domains: *Beliefs about Capabilities* and Beliefs about Consequences. However, *dyspnoea* is also suggested is a somatic and emotional experience in HF. In cardiac disease, when cardiac output and oxygenation is suboptimal breathing is increased to provide appropriate oxygenation. Due to this, the physical capacity to engage in physical activity is reduced and it places a burden on the cardiopulmonary system. This results in *Dyspnoea*–a clinical symptom of shortness of breath. *Dyspnoea* in HF is a threatening and aversive experience. Besides imposing physical restrictions in HF, *dyspnoea* also directly punishes physical activity behaviour (*Reinforcement*: punishment). On the other hand, *dyspnoea* is a fear-inducing somatic experience of HF and can be considered within the fear-avoidance model [50]. In this context, perceived threat associated with *dyspnoea* influence coping with HF and engagement in physical activity. Therefore, three interpretations of the impact of *dyspnoea* on physical activity in stable HF exist: (a) Pavlovian conditioning; (b) operant conditioning [51]; and (c) cognitive-behavioural response to symptoms. While the former two are concerned with habit breaking (i.e., model-free processes), the latter is concerned with the underlying cognitive and behavioural response to somatic experience (i.e., beliefs about symptoms, catastrophising and dyspnoea-related fear-avoidance beliefs).

Because acute *dyspnoea* is associated with a heart attack, physical activity becomes punishing as it induces getting out of breath. Such conditioning can be extinguished through repeated performance of physical activity. During this performance, the somatic state of being out of breath is not followed by pain or hospitalisation, so the pairing is unlearned. Not only punishment following physical activity needs to be lifted, getting out of breath on physical exertion needs to be paired with a positive reward. This happens in habitual physical activity over time when physical activity becomes intrinsically rewarding [52]. However, for habitual physical activity to be established, an individual needs to practice and repeat the behaviour [53]. Individuals living with HF find the experience of getting out of breath aversive enough that they do not engage in the behaviour at all in the first place. When they do persist, over time, their exercise capacity improves and permits them to engage in physical activity without limitations.

## Strength and limitations

The participants recruited for this study are representative of the general population of individuals living with heart failure attending a specialist clinic in terms of their age and clinical and demographic characteristics. However, they may have a better clinical outcome than on average. This sample was hospitalised three times fewer days a year than the general population [54].

This study, using the TDF, was able to elicit a comprehensive list of relevant domains. Belief statements provide the basis for the content development of a scale designed to assess physical activity barriers and enablers in HF. The identified TDF domains suggest key constructs and domains that need to be investigated in a quantitative study. Causal belief statements were

formulated to describe the causal links among the relevant TDF domains as perceived by the participants.

This study systematically described the causal links between relevant factors influencing physical activity in HF. The systematicity was facilitated by clearly defined criteria for inferring causality–lexico-syntactic patterns. This may impose a limitation, as it may have resulted in the omission of the links that are drawn implicitly without apparent lexico-syntactic patterns.

The assessment of the validity of the qualitative findings was carried out [55]. Reflexivity and reflection were ensured throughout the study. A coding scheme, as well as the inter-rater reliability analysis, were carried out to improve reliability of the findings. The ecological validity of the findings was ensured through the recruitment of a representative sample and rich and thick verbatim analysis.

## Clinical implication

According to the findings of this study, clinical advice and reassurance that reduces negative expectancies surrounding physical activity (e.g., secondary heart attack) are the first necessary means to physical activity change in HF.

The findings of this study also recognise that to improve physical activity engagement in HF, one should address the detrimental effect of *dyspnoea* on the behaviour. It is essential to engage an individual in physical activity, gradually increasing the intensity and duration. Once an individual has built the physical capacity to engage in physical activity and has repeatedly been engaging in physical activity, they will experience exercise-induced breathlessness that is not followed by adverse outcomes, which will lead to a new learnt response to exercise-induced breathlessness.

The findings indicate a clear need for an intervention that addresses the following perceived barriers: maladaptive beliefs about the major health-related event that preceded HF (e.g., heart attack, acute decompensation), maladaptive beliefs about implantable devices that lead to fearful avoidance of physical activity, and maladaptive beliefs about breathlessness upon exertion. It is recommended to explore these beliefs in a non-judgemental and open manner, ensuring patient-driven change. Furthermore, an education programme differentiating HF-related dyspnoea from the expected exercise-induced breathlessness is needed (i.e., reassurance in the safety of physical activity even when it induces getting out of breath). Overall, according to the findings of this study, a cognitive behavioural intervention that addresses the fear-avoidance response to breathlessness is deemed helpful for engaging individuals living with HF in physical activity. The common approaches include Cognitive Behavioural Therapy (CBT) and Acceptance and Commitment Therapy, ACT [56, 57]. While CBT can address the maladaptive beliefs about breathlessness and negative emotional response (i.e., anxiety, fear), ACT can promote acceptance, reduce avoidant behaviour triggered by breathlessness and change maladaptive coping.

In addition, based on the findings of this study, it is recommended to engage individuals living with HF in a gradual increase in physical activity intensity, foster their skill mastery and use verbal persuasion, which may help improve physical activity self-efficacy [58]. This study suggests that it is essential to set specific, measurable, achievable, relevant, time-bound physical activity goals and detailed action plans, including implementation intentions (i.e., if it rains, I will walk around the house). Similarly to the study informing REACH-HF intervention development [59], this study found that goal setting and behavioural regulation concerned with self-care behaviours in HF can be facilitated by instrumental and emotional social support. Therefore, it is recommended to set goals and make plans with others who have a similar level of physical capacity to engage in physical activity as those affected by HF. Thus, individuals

living with HF might benefit from a walking group, where participants support each other in making explicit goals and action plans and social contracts to act on them.

It appears evident that to address the barriers and amplify the enablers, a behaviour change intervention is needed. According to the findings of this study, this intervention should closely follow HAPA [48, 49]. In addition, there is a need for explicit clinical advice and reassurance in the safety of physical activity educational programme, CBT and ACT to take place as soon as possible after the major event. It is proposed to investigate the effectiveness and acceptability of such interventions in a RCT and subsequently consider these interventions for inclusion in clinical guidelines.

A broad range of professional expertise is required to support physical activity engagement in HF. Therefore, a multidisciplinary healthcare team comprising psychotherapists, clinical psychologists, HF nurses and cardiologists is needed to provide comprehensive healthcare and improve clinical outcomes in HF.

## Conclusion

The present one-to-one interview study elicited 39 belief statements about barriers and enablers to physical activity in older adults (≥70) living with HF. These belief statements and their pervasiveness across transcripts (i.e., the number of quotes) suggest that breathlessness (i.e., Dyspnoea), *Beliefs about Capabilities*, *Environmental Context and Resources*, *Goal*, *Behavioural Regulation*, *Beliefs about Consequences*, and *Social Influences* are most relevant to physical activity in HF. This study provides insights for cardiologists, HF-specialist nurses, and physiotherapists to help design and deliver a physical activity intervention to individuals living with HF.

## Author Contributions

**Conceptualization:** Aliya Amirova, Martin R. Cowie.

**Data curation:** Aliya Amirova, Rebecca Lucas.

**Formal analysis:** Aliya Amirova.

**Investigation:** Aliya Amirova, Martin R. Cowie, Mark Haddad.

**Methodology:** Aliya Amirova, Mark Haddad.

**Project administration:** Aliya Amirova, Rebecca Lucas.

**Supervision:** Martin R. Cowie, Mark Haddad.

**Validation:** Rebecca Lucas.

**Writing – original draft:** Aliya Amirova.

**Writing – review & editing:** Aliya Amirova, Rebecca Lucas, Martin R. Cowie, Mark Haddad.

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
