## [Decision Letter · Decision Letter 0]

29 Mar 2022

PONE-D-21-37336Perceived barriers and enablers influencing physical activity in heart failure: a qualitative one-to-one interview studyPLOS ONE

Dear Dr. Amirova,

Thank you for submitting your manuscript to PLOS ONE. After careful consideration, we feel that it has merit but does not fully meet PLOS ONE’s publication criteria as it currently stands. Therefore, we invite you to submit a revised version of the manuscript that addresses the points raised during the review process.

Kind regards,

Tim Luckett

Academic Editor

PLOS ONE

Journal Requirements:

Reviewers' comments:

Reviewer's Responses to Questions

**Comments to the Author**

1. Is the manuscript technically sound, and do the data support the conclusions?

Reviewer #1: Yes

Reviewer #2: Yes

2. Has the statistical analysis been performed appropriately and rigorously? 

Reviewer #1: Yes

Reviewer #2: N/A

3. Have the authors made all data underlying the findings in their manuscript fully available?

Reviewer #1: No

Reviewer #2: No

4. Is the manuscript presented in an intelligible fashion and written in standard English?

Reviewer #1: Yes

Reviewer #2: Yes

5. Review Comments to the Author

Reviewer #1: Thank you for the opportunity to review your interesting manuscript. I have made a few comments/suggestions.

First statement should be referenced, should include LVEF and NYHA class discussion here along with symptoms

Second paragraph needs more detail about 1) why is exercise recommended for HF and 2) expand on the complexity and challenges

Fourth paragraph follows on better from second paragraph, perhaps combine them and add more detail about the challenges not just repeat complexities which is vague

Line 104 NYHA class should be spelt out here

Line 185 is that correctly worded? If it has a low frequency of quotes but was mentioned by all the participants would that not be common?

Line 187 how were sedentary versus active individuals identified/defined?

Line 224 This is more a discussion point/result than a participant characteristic, additionally it is referred to as Breathlessness but later on the authors discuss dyspnoea, suggest move and clarify breathlessness/dyspnoea

Table 1 Should there be a heading between ‘Implantable device’ and ‘Treatment’ sections?

Table 2 Order the results by frequency i.e. ‘group programmes’ should come before ‘facilities’ in first section, ‘physical activity improves..’ ahead of ‘I engage in …’ in a later section

The section titled Clinical implications includes parts which would fit better in discussions such as lines 371-379

Reviewer #2: This study uses qualitative interviews to investigate what motivates older patients (>70) to engage in physical activity and what are the barriers to doing so. The study discerned belief statements from the participants about what they perceive to be these enablers and barriers, and analysed these statements using the Theoretical Domains Framework. The statements that demonstrate causal were captured in a model, to demonstrate how behaviour regulation works for HF patients around physical activity: how they come to perform or not perform physical activity. This model is then used to generate clinical recommendations and an intervention that focus on how to address the specific barriers (e.g. breathlessness as a symptom of exercise as opposed to HF, education targeted at specific misunderstandings) that could improve patient activity

The paper exemplifies how qualitative research should be done and written up. It offers a rigorous qualitative analysis that is well-designed and easy to understand. The TDF has been put to good use to interpret the interview data and the authors provide a clear overview of the various enablers and barriers for patients to engage in physical activity. The detailed Methods section could serve as an example for others on how to report qualitative research findings with clear descriptions of the setting and participants, the sampling, and especially data processing and analysis. The section is highly detailed, giving readers all the necessary information about the study, but still concise.

The discussion of the findings is clear in explaining how the domains are revealed in the interviews, giving a clear understanding of the perceptions reported by the research participants. This provides a valuable and solid evidence-base for a to-be developed intervention. As the authors suggest, a quantitative study could focus specifically on the belief statements that this paper discerned, and investigate for a large sample to what extent they apply to the population. This can then be used to develop a targeted and evidence-based intervention. That is, of course, not to say that this paper has value only as a pre-study for a survey or experiment. How participants articulate their beliefs is highly valuable in designing an appropriate intervention and understanding the problems around physical activity.

I don’t think the paper has any major weaknesses. However, I do have some minor questions and suggestions.

1. It would be helpful to have a bit more elaboration in the introduction about what TDF is and why the authors chose to use it. Most readers will not be familiar with it, and given its importance for coding the interviews and generating the analysis, a bit of background would facilitate this understanding.

2. I am curious why the authors chose the specific causal connectors mentioned in Generating Causal Belief Statements. While they acknowledge in the Discussion that there may be others that have been missed, particularly implicit connectors (people often use “and” when making causal connections), that makes the question more salient.

3. I am not entirely clear about the difference between the domains “Engaging in physical activity is (not) a priority for me” and “physical activity is important to me”; particularly the distinction between engaging in physical activity, and physical activity. I can imagine this would have made coding tricky. (Not something that needs to be addressed in the paper, but it’s a question I have after reading the paper.)

4. The exemplar quote in the Goal Domain “I already engage…”; I don’t understand how this is a barrier, unless the point is that it is not enough. But I can imagine that this is tricky. One participant reporting that they are doing all they can may be doing more than enough, whereas another is doing too little. For the second, such a belief clearly is a barrier, but for the former it’s not. How is it coded and analysed here?

5. In the Social Influenced Domain, the quote about other people are role models seems less about role models, and more about social norms. Indeed, the participant says that they themselves have always been very active, which suggests they are not following a role model, but habits or conventions: they are active, because that’s what they’ve always done. With a role model, I would expect that they had become active, because somebody inspired them (which may of course be a family member).

6. Looking at the conclusion that patients should learn that breathlessness is not a problem, I can imagine that in the Behavioural Regulation Domain, the belief “I pace my physical activity…” is actually a barrier, since as the quote shows, as soon as the participant is a bit breathless, they slow down. Which would be counterproductive.

6. PLOS authors have the option to publish the peer review history of their article (what does this mean?). If published, this will include your full peer review and any attached files.

Reviewer #1: No

Reviewer #2: **Yes: **Lucas Seuren

---

## [Author Response · Author response to Decision Letter 0]

5 May 2022

PONE-D-21-37336

Dear Dr Tim Luckett, Reviewer 1, and Dr Lucas Seuren, 

Thank you for reviewing our manuscript: "Perceived barriers and enablers influencing physical activity in heart failure: a qualitative one-to-one interview study". 

Many thanks to you for these insightful and helpful comments. We have now made changes to the manuscript in response to these reviewers’ comments as outlined below. 

Thank you, we look forward to hearing from you. 

Sincerely, 

Aliya on behalf of co-authors: 

Rebecca Lucas, Professor Martin R Cowie, and Dr Mark Haddad

Response to reviewers' comments: 

Reviewer #1:

Thank you for the opportunity to review your interesting manuscript. I have made a few comments/suggestions.

1. First statement should be referenced, should include LVEF and NYHA class discussion here along with symptoms

Thank you. We now added lines 47-52: 

It is characterised by signs of volume overload, which may include peripheral oedema and pulmonary rales, and symptoms of breathlessness (dyspnoea), effort intolerance and fatigue. The extent to which these symptoms limit physical activity can be categorised using New York Heart Association (NYHA) classification (class I – no limitations; Class II – symptoms on ordinary exertion; Class III – symptoms on less than ordinary exertion; and Class IV – symptoms at rest) (1). Traditionally, two main “types” of HF exist, heart failure with reduced left ventricular ejection fraction (HFrEF) and heart failure with preserved LVEF (HFpEF) (2).

2. Second paragraph needs more detail about 1) why is exercise recommended for HF and 2) expand on the complexity and challenges

We now added rationale for why physical activity is recommended to HF patients on line 56-57: 

Physical activity is associated with improved quality of life (3–5), reduced hospitalisation (4), and increased longevity (6). A physically active lifestyle, therefore, is a component of the recommended treatment (2).

We also described the challenges HF patients face when engaging in physical activity (lines 61-80): 

Several of the barriers to attending CR are speculated to exist at the healthcare system and broader socio-economic level (such as availability of cardiac rehabilitation programmes and exercise programmes, transportation issues, low referral rates, and limited knowledge of available programmes), as outlined in a consensus paper on adherence to exercise programmes and its barriers (7). A recent systematic review identified that older age, depression and low LVEF are considerable barriers to everyday physical activity engagement in HF patients (8). There is moderate evidence in support of the following modifiable barriers – symptom distress, negative emotional response towards physical activity – and modifiable enablers – social support, self-efficacy, and positive attitude towards physical activity. However, research into modifiable barriers and enablers is limited (8). 

Fourth paragraph follows on better from second paragraph, perhaps combine them and add more detail about the challenges not just repeat complexities which is vague. 

Thank you, we moved paragraphs accordingly. 

Line 104 NYHA class should be spelt out here

The NYHA was spelt out on line 113 in this version. 

Line 185 is that correctly worded? If it has a low frequency of quotes but was mentioned by all the participants would that not be common?

We now rephrased this on lines 192-193: 

"either relatively pervasive (i.e., high number of quotes, k) or common (expressed by a large proportion of participants, n), or both. For a belief to be considered relevant, it had to be expressed by more than two participants at least."

Line 187 how were sedentary versus active individuals identified/defined?

We now added lines 239-241: 

From self-reported within transcripts descriptions of everyday physical activity, participants were categorised into (i) sedentary individuals and those performing (ii) moderate physical activity and (iii) vagarious physical activity at least once a week. 

In the results section, line 274-281, we added: 

Sedentary behaviour (n=1): Participant 5 was mostly sedentary; however, he used an exercise step once a week as prescribed by nurse and walked indoors.

 Moderate physical activity (n=15): All but Participant 5 walked daily. Two participants walked for at least 30 minutes as part of everyday activities of daily living (Participants 3 and 4). Other participants walked for leisure in addition to activities of daily living at least once a week. 

 Vigorous physical activity (n=8): Three participants had an exercise routine (Participants 7, 11, 8); two attended a gym (Participants 2 and 1), one – a sports club (bowls; Participant 15), one – an aqua-aerobics class (Participant 1), and one participant exercised regularly using a rowing machine (Participant 16).

Line 224 This is more a discussion point/result than a participant characteristic, additionally it is referred to as Breathlessness but later on the authors discuss dyspnoea, suggest move and clarify breathlessness/dyspnoea

Apologies, the subheading "Findings" was missing in the previous version, and it appeared that Breathlessness findings were reported as part of "Participants Characteristics". 

Table 2 Order the results by frequency i.e. 'group programmes' should come before 'facilities' in first section, 'physical activity improves..' ahead of 'I engage in …' in a later section

In Table 2, we reordered all beliefs by the number of quotes supporting them. 

The section titled Clinical implications includes parts which would fit better in discussions such as lines 371-379

Apologies for the confusion, we meant for this to be presented as part of a behaviour change intervention or psychotherapy recommendation. We now rephrased accordingly on lines 446-480: 

The findings indicate a clear need for an intervention that addresses the following perceived barriers: maladaptive beliefs about the major health-related event that preceded HF (e.g., heart attack, acute decompensation), maladaptive beliefs about implantable devices that lead to fearful avoidance of physical activity, and maladaptive beliefs about breathlessness upon exertion. It is recommended to explore these beliefs in a non-judgemental and open manner, ensuring patient-driven change. Furthermore, an education programme differentiating HF-related dyspnoea from the expected exercise-induced breathlessness is needed (i.e., reassurance in the safety of physical activity even when it induces getting out of breath). Overall, according to the findings of this study, a cognitive behavioural intervention that addresses the fear-avoidance response to breathlessness is deemed helpful for engaging individuals living with HF in physical activity. The common approaches include Cognitive Behavioural Therapy (CBT) and Acceptance and Commitment Therapy, ACT (9,10). While CBT can address the maladaptive beliefs about breathlessness and negative emotional response (i.e., anxiety, fear), ACT can promote acceptance, reduce avoidant behaviour triggered by breathlessness and change maladaptive coping. 

In addition, based on the findings of this study, it is recommended to engage individuals living with HF in a gradual increase in physical activity intensity, foster their skill mastery and use verbal persuasion, which may help improve physical activity self-efficacy (11). This study suggests that it is essential to set specific, measurable, achievable, relevant, time-bound physical activity goals and detailed action plans, including implementation intentions (i.e., if it rains, I will walk around the house). Similarly to the study informing REACH-HF intervention development (12), this study found that goal setting and behavioural regulation concerned with self-care behaviours in HF can be facilitated by instrumental and emotional social support. Therefore, it is recommended to set goals and make plans with others who have a similar level of physical capacity to engage in physical activity as those affected by HF. Thus, individuals living with HF might benefit from a walking group, where participants support each other in making explicit goals and action plans and social contracts to act on them. 

It appears evident that to address the barriers and amplify the enablers, a behaviour change intervention is needed. According to the findings of this study, this intervention should closely follow HAPA (13,14). In addition, there is a need for explicit clinical advice and reassurance in the safety of physical activity educational programme, CBT and ACT to take place as soon as possible after the major event. It is proposed to investigate the effectiveness and acceptability of such interventions in an RCT and subsequently consider these interventions for inclusion in clinical guidelines. 

A broad range of professional expertise is required to support physical activity engagement in HF. Therefore, a multidisciplinary healthcare team comprising psychotherapists, clinical psychologists, HF nurses and cardiologists is needed to provide comprehensive healthcare and improve clinical outcomes in HF. 

 

Reviewer #2: 

This study uses qualitative interviews to investigate what motivates older patients (>70) to engage in physical activity and what are the barriers to doing so. The study discerned belief statements from the participants about what they perceive to be these enablers and barriers, and analysed these statements using the Theoretical Domains Framework. The statements that demonstrate causal were captured in a model, to demonstrate how behaviour regulation works for HF patients around physical activity: how they come to perform or not perform physical activity. This model is then used to generate clinical recommendations and an intervention that focus on how to address the specific barriers (e.g. breathlessness as a symptom of exercise as opposed to HF, education targeted at specific misunderstandings) that could improve patient activity

The paper exemplifies how qualitative research should be done and written up. It offers a rigorous qualitative analysis that is well-designed and easy to understand. The TDF has been put to good use to interpret the interview data and the authors provide a clear overview of the various enablers and barriers for patients to engage in physical activity. The detailed Methods section could serve as an example for others on how to report qualitative research findings with clear descriptions of the setting and participants, the sampling, and especially data processing and analysis. The section is highly detailed, giving readers all the necessary information about the study, but still concise.

The discussion of the findings is clear in explaining how the domains are revealed in the interviews, giving a clear understanding of the perceptions reported by the research participants. This provides a valuable and solid evidence-base for a to-be developed intervention. As the authors suggest, a quantitative study could focus specifically on the belief statements that this paper discerned, and investigate for a large sample to what extent they apply to the population. This can then be used to develop a targeted and evidence-based intervention. That is, of course, not to say that this paper has value only as a pre-study for a survey or experiment. How participants articulate their beliefs is highly valuable in designing an appropriate intervention and understanding the problems around physical activity.

I don't think the paper has any major weaknesses. However, I do have some minor questions and suggestions.

1. It would be helpful to have a bit more elaboration in the introduction about what TDF is and why the authors chose to use it. Most readers will not be familiar with it, and given its importance for coding the interviews and generating the analysis, a bit of background would facilitate this understanding.

We now described and justified the use of TDF on lines 98-120: 

The use of TDF offers the following advantages over the IMF. The TDF offers means to systematise and structure qualitative analysis. Theoretical Domains Framework (15) is a tool developed through an international collaborative effort. It systematically describes domains and constructs which influence behaviour under investigation (16). TDF summarises constructs of existing behaviour change theories into 14 domains, such as Knowledge, Skills, Social/Professional Role and Identity, Beliefs about Capabilities, Optimism, Beliefs about Consequences, Reinforcement, Intentions, Goals; Memory, Attention and Decision Processes; Environmental Context and Resources; Social influences; Emotion; and Behavioural Regulation. Domains. TDF has been widely used in research (16), including research on physical activity in healthy adults (17,18). It has been suggested that TDF-based semi-structured interview helps in eliciting a greater number of relevant barriers and enablers, in contrast to unstructured interviews or less structured interviews which are likely to result in the identification of only some, usually the most salient, barriers and enablers (19). Thus, TDF-based semi-structured interviews are expected to facilitate the search for a broader range of domains that are perceived as relevant to physical activity by individuals living with HF.

2. I am curious why the authors chose the specific causal connectors mentioned in Generating Causal Belief Statements. While they acknowledge in the Discussion that there may be others that have been missed, particularly implicit connectors (people often use "and" when making causal connections), that makes the question more salient.

Thank you for your feedback. We were guided by literature in using these connectors. These clearly-defined criteria were followed to ensure that the transcripts were analysed systematically. This helped in ensuring that causal beliefs were formulated from quotes that were systematically extracted from the transcripts, as well as in ensuring that enough evidence in the transcripts was present to infer that the belief is causal. We followed the literature cited below that described and justified these lexical patterns as indicative of causal inference: 

Comrie B. The syntax of causative constructions: cross-language similarities and divergences. The grammar of causative constructions. 1976;259–312. 

Roese NJ, Olson JM. Counterfactual thinking: A critical overview. What might have been: The social psychology of counterfactual thinking. 1995;1–55. 

Tversky A, Kahneman D. Judgment under Uncertainty: Heuristics and Biases. Science. 1974 Sep 27;185(4157):1124–31. 

Sasaki, S., Takase, S., Inoue, N., Okazaki, N., & Inui, K. (2017). Handling multiword expressions in causality estimation. In IWCS 2017—12th International Conference on Computational Semantics—Short papers.

3. I am not entirely clear about the difference between the domains "Engaging in physical activity is (not) a priority for me" and "physical activity is important to me"; particularly the distinction between engaging in physical activity, and physical activity. I can imagine this would have made coding tricky. (Not something that needs to be addressed in the paper, but it's a question I have after reading the paper.)

Thank you – this is a good point - we now merged these two beliefs into one: "physical activity is a priority for me". 

4. The exemplar quote in the Goal Domain "I already engage…"; I don't understand how this is a barrier, unless the point is that it is not enough. But I can imagine that this is tricky. One participant reporting that they are doing all they can be doing more than enough, whereas another is doing too little. For the second, such a belief clearly is a barrier, but for the former it's not. How is it coded and analysed here?

The belief reflects the fact that HF patients perceived to be engaging in enough physical activity, while from their self-report it was evident that they did not. Physical activity levels are now reported on lines 267-274. 

5. In the Social Influenced Domain, the quote about other people are role models seems less about role models, and more about social norms. Indeed, the participant says that they themselves have always been very active, which suggests they are not following a role model, but habits or conventions: they are active, because that's what they've always done. With a role model, I would expect that they had become active, because somebody inspired them (which may of course be a family member).

Thank you, this is a really helpful reviewer insight. Upon the consideration of this comment, we have revised the labelling of this belief. We checked the quotes and realised that this quote referred by the reviewer, as well as other associated quotes supporting this belief refer to social norm rather than "role models". Therefore, we now have changed the role model belief into: "I am physically active because everyone I know closely is physically active", and changed the construct from “social modelling” to “social norm” (Table 2). 

6. Looking at the conclusion that patients should learn that breathlessness is not a problem, I can imagine that in the Behavioural Regulation Domain, the belief "I pace my physical activity…" is actually a barrier, since as the quote shows, as soon as the participant is a bit breathless, they slow down. Which would be counterproductive.

Thank you, we now added the clarification on this finding. In this instance, individuals were pacing physical activity and intensity to make sure that they engage in a prolonged period of physical activity, rather than very short bursts of high-intensity exercise beyond one's physical capacity or bursts of activity that brought on symptoms, patients were using pacing as a strategy to avoid sudden onset of extreme exertion. 

2. In your Data Availability statement, you have not specified where the minimal data set underlying the results described in your manuscript can be found. PLOS defines a study's minimal data set as the underlying data used to reach the conclusions drawn in the manuscript and any additional data required to replicate the reported study findings in their entirety. All PLOS journals require that the minimal data set be made fully available. For more information about our data policy, please see http://journals.plos.org/plosone/s/data-availability. Upon re-submitting your revised manuscript, please upload your study's minimal underlying data set as either Supporting Information files or to a stable, public repository and include the relevant URLs, DOIs, or accession numbers within your revised cover letter. For a list of acceptable repositories, please see http://journals.plos.org/plosone/s/data-availability#loc-recommended-repositories. Any potentially identifying patient information must be fully anonymized. Important: If there are ethical or legal restrictions to sharing your data publicly, please explain these restrictions in detail. Please see our guidelines for more information on what we consider unacceptable restrictions to publicly sharing data: http://journals.plos.org/plosone/s/data-availability#loc-unacceptable-data-access-restrictions. Note that it is not acceptable for the authors to be the sole named individuals responsible for ensuring data access. We will update your Data Availability statement to reflect the information you provide in your cover letter.

In response to the reviewer's comments the following references were added: 

The Criteria Committee of the New York Heart Association. (1994). Nomenclature and Criteria for Diagnosis of Diseases of the Heart and Great Vessels (9th ed.). Boston: Little, Brown & Co. pp. 253–256.

Taylor RS, Long L, Mordi IR, Madsen MT, Davies EJ, Dalal H, et al. Exercise-Based Rehabilitation for Heart Failure: Cochrane Systematic Review, Meta-Analysis, and Trial Sequential Analysis. JACC Heart Fail. 2019 Jul 10;7(8):691–705. 

Sagar VA, Davies EJ, Briscoe S, Coats AJ, Dalal HM, Lough F, et al. Exercise-based rehabilitation for HF: systematic review and meta-analysis. Open heart. 2015;2:1. 

Lewinter C, Doherty P, Gale CP, Crouch S, Stirk L, Lewin RJ, et al. Exercise-based cardiac rehabilitation in patients with heart failure: a meta-analysis of randomised controlled trials between 1999 and 2013. Eur J Prev Cardiol. 2015 Dec;22(12):1504–12. 

Belardinelli R, Georgiou D, Cianci G, Purcaro A. 10-year exercise training in chronic heart failure: a randomized controlled trial. J Am Coll Cardiol. 2012 Oct 16;60(16):1521–8. 

Amirova A, Taylor L, Volkmer B, Ahmed N, Chater A, Fteropoulli T. Informing behaviour change intervention design using Bayesian meta-analysis: physical activity in heart failure. Medrxiv. 2021 Sep 1. (Under review at Health Psychology Review: second round)

McDonald S, O'Brien N, White M, Sniehotta FF. Changes in physical activity during the retirement transition: a theory-based, qualitative interview study. Int J Behav Nutr Phys Act. 2015 Feb 21;12:25. 

Taylor N, Lawton R, Conner M. Development and initial validation of the determinants of physical activity questionnaire. International Journal of Behavioral Nutrition and Physical Activity. 2013;10(1):74. 

Francis JJ, Stockton C, Eccles MP, Johnston M, Cuthbertson BH, Grimshaw JM, et al. Evidence-based selection of theories for designing behaviour change interventions: using methods based on theoretical construct domains to understand clinicians' blood transfusion behaviour. Br J Health Psychol. 2009 Nov;14(Pt 4):625–46.

 

References

1. The Criteria Committee of the New York Heart Association. Nomenclature and Criteria for Diagnosis of Diseases of the Heart and GreatVessels . 9th ed. New Yor Heart Association TCC of the NYHA, editor. Boston: Little, Brown & Co; 1994. 

2. National Institute for Healthcare and Excellence. NICE: Overview. Chronic heart failure in adults: diagnosis and management. Guidance [Internet]. Overview. Chronic heart failure in adults: diagnosis and management. Guidance. 2018 [cited 2020 Mar 23]. Available from: https://www.nice.org.uk/guidance/ng106

3. Taylor RS, Long L, Mordi IR, Madsen MT, Davies EJ, Dalal H, et al. Exercise-Based Rehabilitation for Heart Failure: Cochrane Systematic Review, Meta-Analysis, and Trial Sequential Analysis. JACC Heart Fail. 2019 Jul 10;7(8):691–705. 

4. Sagar VA, Davies EJ, Briscoe S, Coats AJ, Dalal HM, Lough F, et al. Exercise-based rehabilitation for HF: systematic review and meta-analysis. Open heart. 2015;2:1. 

5. Lewinter C, Doherty P, Gale CP, Crouch S, Stirk L, Lewin RJ, et al. Exercise-based cardiac rehabilitation in patients with heart failure: a meta-analysis of randomised controlled trials between 1999 and 2013. Eur J Prev Cardiol. 2015 Dec;22(12):1504–12. 

6. Belardinelli R, Georgiou D, Cianci G, Purcaro A. 10-year exercise training in chronic heart failure: a randomized controlled trial. J Am Coll Cardiol. 2012 Oct 16;60(16):1521–8. 

7. Conraads VM, Deaton C, Piotrowicz E, Santaularia N, Tierney S, Piepoli MF, et al. Adherence of heart failure patients to exercise: barriers and possible solutions: a position statement of the Study Group on Exercise Training in Heart Failure of the Heart Failure Association of the European Society of Cardiology. Eur J Heart Fail. 2012 May;14(5):451–8. 

8. Amirova A, Taylor L, Volkmer B, Ahmed N, Chater A, Fteropoulli T. Informing behaviour change intervention design using Bayesian meta-analysis: physical activity in heart failure. Medrxiv. 2021 Sep 1; 

9. Ivanova E, Jensen D, Cassoff J, Gu F, Knäuper B. Acceptance and commitment therapy improves exercise tolerance in sedentary women. Med Sci Sports Exerc. 2015 Jun;47(6):1251–8. 

10. Hayes SC, Bissett RT, Korn Z, Zettle RD, Rosenfarb IS, Cooper LD, et al. The impact of acceptance versus control rationales on pain tolerance. Psychol Rec. 1999 Jan;49(1):33–47. 

11. Bandura A. Self-efficacy: The exercise of control: Macmillan. 1997. 

12. Greaves CJ, Wingham J, Deighan C, Doherty P, Elliott J, Armitage W, et al. Optimising self-care support for people with heart failure and their caregivers: development of the Rehabilitation Enablement in Chronic Heart Failure (REACH-HF) intervention using intervention mapping. Pilot Feasibility Stud. 2016 Aug 2;2:37. 

13. Schwarzer R. Health action process approach (HAPA) as a theoretical framework to understand behavior change. AP. 2016 Dec 5;30(121):119. 

14. Zhang C-Q, Zhang R, Schwarzer R, Hagger MS. A meta-analysis of the health action process approach. Health Psychol. 2019 Jul;38(7):623–37. 

15. Cane J, O’Connor D, Michie S. Validation of the theoretical domains framework for use in behaviour change and implementation research. Implement Sci. 2012 Apr 24;7:37. 

16. Atkins L, Francis J, Islam R, O’Connor D, Patey A, Ivers N, et al. A guide to using the Theoretical Domains Framework of behaviour change to investigate implementation problems. Implement Sci. 2017 Jun 21;12(1):77. 

17. McDonald S, O’Brien N, White M, Sniehotta FF. Changes in physical activity during the retirement transition: a theory-based, qualitative interview study. Int J Behav Nutr Phys Act. 2015 Feb 21;12:25. 

18. Taylor N, Lawton R, Conner M. Development and initial validation of the determinants of physical activity questionnaire. International Journal of Behavioral Nutrition and Physical Activity. 2013;10(1):74. 

19. Francis JJ, Stockton C, Eccles MP, Johnston M, Cuthbertson BH, Grimshaw JM, et al. Evidence-based selection of theories for designing behaviour change interventions: using methods based on theoretical construct domains to understand clinicians’ blood transfusion behaviour. Br J Health Psychol. 2009 Nov;14(Pt 4):625–46.

---

## [Decision Letter · Decision Letter 1]

7 Jul 2022

Perceived barriers and enablers influencing physical activity in heart failure: a qualitative one-to-one interview study

PONE-D-21-37336R1

Dear Dr. Amirova,

We’re pleased to inform you that your manuscript has been judged scientifically suitable for publication and will be formally accepted for publication once it meets all outstanding technical requirements.

Kind regards,

Tim Luckett

Academic Editor

PLOS ONE

Reviewers' comments:

Reviewer's Responses to Questions

**Comments to the Author**

1. If the authors have adequately addressed your comments raised in a previous round of review and you feel that this manuscript is now acceptable for publication, you may indicate that here to bypass the “Comments to the Author” section, enter your conflict of interest statement in the “Confidential to Editor” section, and submit your "Accept" recommendation.

Reviewer #2: All comments have been addressed

2. Is the manuscript technically sound, and do the data support the conclusions?

Reviewer #2: Yes

3. Has the statistical analysis been performed appropriately and rigorously? 

Reviewer #2: N/A

4. Have the authors made all data underlying the findings in their manuscript fully available?

Reviewer #2: No

5. Is the manuscript presented in an intelligible fashion and written in standard English?

Reviewer #2: Yes

6. Review Comments to the Author

Reviewer #2: The first submitted version of the paper provided a clear analysis of of motivation in older HF patients on whether to exercise. The minor comments raised by myself and Reviewer #1 have been more than adequately addressed. I want to thank the authors for engaging with the feedback in such a clear an open-minded way, and I apologize for the delay in reviewing the final manuscript.

7. PLOS authors have the option to publish the peer review history of their article (what does this mean?). If published, this will include your full peer review and any attached files.

Reviewer #2: **Yes: **Lucas Seuren

---

## [Editor Report · Acceptance letter]

20 Jul 2022

PONE-D-21-37336R1 

Perceived barriers and enablers influencing physical activity in heart failure: a qualitative one-to-one interview study 

Dear Dr. Amirova:

I'm pleased to inform you that your manuscript has been deemed suitable for publication in PLOS ONE. Congratulations! Your manuscript is now with our production department. 

Kind regards, 

on behalf of

Dr. Tim Luckett 

Academic Editor

PLOS ONE